# Isolation, Identification, and Evaluation of the Predatory Activity of Chinese *Arthrobotrys* Species towards Economically Important Plant-Parasitic Nematodes

**DOI:** 10.3390/jof9121125

**Published:** 2023-11-21

**Authors:** Yan Wu, Zaifu Yang, Zhaochun Jiang, Mir Muhammad Nizamani, Hui Zhang, Mingrui Liu, Shan Wei, Yong Wang, Kaihuai Li

**Affiliations:** 1Department of Plant Pathology, College of Agriculture, Guizhou University, Guiyang 550025, China; 18164816527m@sina.cn (Y.W.); mirmohammadnizamani@outlook.com (M.M.N.); huizhang0909@126.com (H.Z.); mingruiliur@163.com (M.L.); swei@gzu.edu.cn (S.W.); yongwangbis@aliyun.com (Y.W.); likh@gzu.edu.cn (K.L.); 2Institute of Vegetable Industry Technology Research, Guizhou University, Guiyang 550025, China; 3Guizhou Station of Plant Protection and Quarantine, Guiyang 550001, China; jiang1zc@sina.com

**Keywords:** identification, predatory activity, *Arthrobotrys*, plant-parasitic nematode

## Abstract

The current investigation aimed to isolate and identify predatory fungal strains and evaluate their efficacy in mitigating the effects of plant-parasitic nematodes. We successfully isolated three distinct nematophagous fungal strains from soil samples, identified as *Arthrobotrys megalosporus*, *A. oligospora*, and *A. sinensis*, using conventional and molecular identification methodologies. In vitro trials illustrated the high capture efficiency of these fungi against plant-parasitic nematodes. Over an exposure period of 48 h to *Aphelenchoides besseyi*, *Bursaphelenchus xylophilus*, and *Ditylenchus destructor*, *A. megalosporus* (GUCC220044) displayed predation rates of 99.7%, 83.0%, and 21.1%, respectively. *A. oligospora* (GUCC220045) demonstrated predation rates of 97.3%, 97.3%, and 54.6%, and *A. sinensis* (GUCC220046) showed rates of 85.1%, 68.3%, and 19.0% against the same cohort of nematodes. The experimental outcomes substantiate that all three identified fungal strains demonstrate predatory activity against the tested nematodes, albeit with varying efficiencies.

## 1. Introduction

Plant-parasitic nematodes, comprising approximately 4300 species, account for around 15% of all nematode families and are critical pathogens causing global economic losses of up to $1500 billion per annum [1,2,3]. These nematodes exhibit a broad host range, including staple crops, vegetables, and fruits [4]. These nematodes primarily infiltrate plants via the root system, leading to chlorotic yellowing, wilting, and, in severe cases, plant death during intermediate-to-late stages of infection. As such, plant nematodes pose a significant threat to agricultural and forestry safety and hinder their development [5].

Traditional nematode disease control methods encompass agricultural and physical control strategies, such as field sanitation, crop rotation, and soil tillage [6]. Additionally, utilizing nematode-resistant crop varieties constitutes one of the most efficient, economical, and eco-friendly methods to curtail yield losses attributed to nematode diseases [7]. However, due to the scarcity of necessary resistance sources for variety development, commercially viable resistant varieties may not be available for all crops or may only be available for a limited number of specific crops, and they cannot be broadly scaled up [8]. Chemical nematicides remain a vital control measure for preventing and curbing diseases caused by plant nematodes, as they rapidly reduce soil nematode populations to avert substantial economic losses [9]. However, intensifying single-crop cultivation, increasing replanting rates, and escalating pesticide use often exceed the soil’s inherent resilience, resulting in excessive pesticide residues, soil compaction, and deterioration of soil tillage quality [10,11]. Additionally, these strategies are associated with drawbacks that include the development of resistance in parasites, extermination of beneficial soil microorganisms, environmental contamination, and potential health implications [12,13,14].

With rising living standards, increased public concern over food safety has made pesticide residue one of the benchmarks for assessing food quality. Consequently, to protect the environment and minimize pesticide residues, increasing research focus is being directed towards biopesticides—biologically-sourced preparations that employ plants, animals, microorganisms, or secondary metabolites to control diseases, pests, and weeds. Biopesticides offer environmental compatibility, minimal residue, and potential for improving soil environments and promoting plant growth and disease resistance [15].

Soils, as rich microbial habitats, harbor numerous natural enemies of nematodes, including mites, bacteria, fungi, and viruses [16]. Nematophagous fungi, in particular, are considered natural antagonists that participate in farmland ecosystem food chains as decomposers [17]. In response to the presence of nematodes, these fungi modify their hyphal structures into specialized rings or other forms of traps, transforming into predators that prey on nematodes [18]. Nematophagous or endophytic fungi can directly attack, kill, immobilize, or repel nematodes, disrupt their host-seeking behavior, interfere with giant cell development, compete for resources, or employ a combination of these strategies [19]. Upon capturing a nematode, the fungal infection process consists of four steps: degradation of the nematode’s body wall, immobilization of the nematode, invasion by hyphae, and digestion and absorption [20]. Following successful capture, the fungi can secrete various enzymes to break down the nematode’s body wall, infiltrate the body, and acquire nutrients necessary for their own growth [21].

*Arthrobotrys* is a genus first described and named by Corda in 1839, and Zopf (1888) described in detail the unique phenomenon of *A. oligospora* producing adhesion networks to capture nematodes [22]. *Arthrobotrys* species are broadly distributed in nature, particularly in soil, and produce a variety of trap morphologies, ranging from simple adhesive hyphae to complex adhesive trapping structures [23]. Some species are known to immobilize nematodes remotely, indicating their capability to secrete specific nematotoxic metabolites [24]. Current studies on the predation of plant-parasitic nematodes by *Arthrobotrys* species mainly focus on root-knot nematodes (*Meloidogyne* spp.), root lesion nematodes (*Pratylenchus* spp.), and garlic stem nematodes (*Ditylenchus dipsaci*) [25].

In this study, we collected soil samples from three counties within Bijie City, Guizhou Province, from which we isolated forty fungal strains. We identified three of these strains and evaluated their in vitro control efficacy against plant nematodes. Utilizing *A. besseyi*, *B. xylophilus*, and *D. destructor*—three types of plant-parasitic nematodes inhabiting distinct environments—we conducted in vitro experiments to compare the predation effects of the isolated strains on these nematodes.

## 2. Materials and Methods

### 2.1. Nematode Cultures

*B. xylophilus* was provided by the Center for Research and Development of Fine Chemicals, Guizhou University. It was propagated by inoculating *Botrytis cinerea* (this strain was provided by the Laboratory of Plant Pathology, Guizhou University) on Potato Dextrose Agar (PDA) plates, followed by incubation in the dark at 28 °C for seven days [26]. Subsequently, *B. xylophilus* nematodes (approximately 50 individuals) were added into *B. cinerea* plates, and the plates were incubated in the dark at 28 °C for an additional 10 days to facilitate nematode multiplication. The nematodes on the dish lid were rinsed with sterile water, and the nematode suspension was transferred into a 1.5 mL centrifuge tube. After centrifugation and washing three times at 5000 rpm, the nematode suspension was diluted to achieve a concentration of 1000 individuals/mL.

*A. besseyi* samples, identified by Zaifu Yang, were collected from rice plants in Dushan County, Guizhou Province, China, while *D. destructor* samples were provided by the Nematode Laboratory, Fujian Agricultural and Forestry University. These nematodes were propagated on carrot calluses, as per the methodology described by Tülek et al. [27]. Following propagation, nematodes were removed from the carrot callus cultures and sterilized using 0.1% streptomycin sulfate for 10 min. They were washed thrice with double-distilled water and subsequently cultured on carrot discs at 28 °C for 30 days. The nematodes on the carrot calluses were rinsed with sterile water and subsequently transferred into a 1.5 mL centrifuge tube to prepare the nematode suspension. Following three rounds of centrifugation and washing at 5000 rpm, the nematode suspension was diluted to achieve a concentration of 1000 individuals/mL.

### 2.2. Sample Collection and Isolation of Strains

Ten soil samples were procured from the Bijie region of Guizhou and stored at 4 °C in the laboratory. Samples were collected from a soil depth of 5–10 cm [28]. The isolation of fungi was carried out using a direct soil-scattering method, employing *B. xylophilus* nematodes as bait to stimulate the production of nematode fungi. Between 0.5 and 1 g of soil was sprinkled directly onto 2% water agar (WA) plates, taking care to ensure that the soil was not too densely spread, and 1 mL of nematode suspension was added, containing 1000 nematodes [29]. The plates were then incubated at 28 °C, and the presence of predation was observed under a microscope every day. After 7 days of incubation in the water agar plates sprinkled with soil, we could observe nematodes being trapped by the fungus-produced rings, and conidia were growing around the deceased nematodes. The structures producing spores grew upright and produced conidia at their apices. We used a sterilized toothpick to pick hyphae or spores in the vicinity of predated nematodes and transfer them to PDA plates, and we conducted 2–3 purifications. The strains were purified and stored in the refrigerator at 4 °C. Three strains from the isolated ones were chosen for identification and evaluation of their nematode control effects.

### 2.3. Morphological Observations

The fungal strains were inoculated on PDA and cultivated for 7 d at 28 °C. Colony morphology was documented using a stereomicroscope (VHX-7000 digital microscope, Keyence, Japan). A compound light microscope (Axio Scope 5, Carl Zeiss Suzhou Co., Ltd., Suzhou, China) equipped with an Axio Cam 208 color camera (Carl Zeiss Suzhou Co., Ltd., Suzhou, China) was used to photograph and measure conidiophores, conidia, and predation structures.

### 2.4. DNA Extraction, PCR, and Sequencing

For the purpose of DNA extraction, the fungal isolates were cultivated on PDA at a temperature of 28 °C. Roughly 100 mg of mycelium was transferred into a 2 mL microcentrifuge tube with a screw cap containing Lysing Matrix C (provided by MP Biomedicals, Shanghai, China). The DNA was extracted using a rapid fungal genomic DNA isolation kit supplied by Sangon Biotech Company, Limited, Shanghai, China.

The Internal Transcribed Spacer (ITS) and Translation Elongation Factor (TEF) regions were amplified using primer pairs ITS4, ITS5 and 526F, 1567R, respectively [30,31,32]. The Polymerase Chain Reaction (PCR) amplification was conducted as follows: 4 min of initial denaturation at 94 °C, followed by 35 cycles consisting of 45 s of denaturation at 94 °C, 1 min of annealing at 52 °C for ITS and 55 °C for TEF, and 1 min of extension at 72 °C. This was concluded with a final extension for 10 min at 72 °C.

The PCR products were purified and sequenced by Sangon Biotech Company, Limited, Shanghai, China. The sequences generated through this study were deposited into the GenBank database maintained by the National Center for Biotechnology Information (NCBI; available at https://www.ncbi.nlm.nih.gov/; accessed on 26 February 2022).

### 2.5. Phylogenetic Analysis

The genes were aligned using the online program MAFFT (MAFFT alignment and NJ/UPGMA phylogeny (https://mafft.cbrc.jp/alignment/server/index.html; accessed on 26 February 2022)) and manually adjusted using BioEdit 7.5.0.3. These were then linked with the Sequence Matrix. All reliable ITS and TEF sequences of *Arthrobotrys* taxa were downloaded from the GenBank database (Table 1). *Dactylellina mammillata* CBS229.54 and *D. yushanensis* CGMCC3.19713 were selected as outgroups. Phylogenetic trees were inferred via the maximum likelihood (ML) approach.

### 2.6. In Vitro Predatory Activity of the Fungal Isolates against Nematodes

The predation efficacy of the isolated strains was assessed using *B. xylophilus* (pine wood nematodes), *A. besseyi* (rice dry-tip nematodes), and *D. destructor* (potato tuber nematode, potato rot nematode). The isolated strains were cultured on PDA plates for a period of 5 days, after which a mycelial plug with a diameter of 5 mm was excised from the edge of the fungal isolate and transferred to the center of a 2% WA plate. The plate was subsequently incubated at 28 °C for a duration of 7 days.

Post-incubation, 1 mL of nematode suspension, containing 1000 nematodes, was added to the plate. This nematode suspension was uniformly distributed within the periphery of the fungal colonies. A nematode-free 2% WA plate was employed as a control.

At 24 h and 48 h intervals, the predation structures of the tested strains were observed and documented with the assistance of a stereomicroscope. The quantity of deceased nematodes was also recorded. These procedures provided data on the efficacy of the tested strains in predating nematodes.

## 3. Results

### 3.1. Identification of the Fungal Isolates

*A. megalosporus* (GUCC220044) soil samples were sourced from Jinzhong Town, Weining County, Bijie City, Guizhou Province. The purified strains were cultured on PDA plates at 28 °C for 10 days, with the colonies growing to a diameter of 9 cm. The colony color was initially white, producing a pink pigment or no pigment later on. The mycelium was fluffy and woolly. The mycelium was transparent, septate, and had an average width of 3.2 μm. The conidiophores grew erect, with conidia being fusiform, elongate-ellipsoidal, or obovoid, and they had 2–4 septa, usually 2 or 3. Very few spores consisted of four septa. The conidia measured 24.3–41.1 × 11.7–17.5 μm (Figure 1, Table 2). These morphological characteristics were consistent with the description of *A. megalosporus* by Kano et al. [36] but there were some differences in spore size.

*A. oligospora* (GUCC220045) samples were randomly collected from terrestrial soil in Dafang County, Bijie City, Guizhou Province. The purified strain was incubated at 28 °C for 7 days and the mycelium grew over the entire 9 cm Petri dish. The colonies were sparse and cottony, with the colonies gradually turning from white to light yellow. The backside of the colonies produced a yellow pigment. On microscopic inspection, the mycelium surface was smooth, branched, and septate. The conidiophore was transparent, grew upright, and had an expansion site producing conidia. The conidiophore continued to grow, producing new expansion sites and generating spores following a repeat of the previous growth mode. The conidia are pyriform or obovoid in shape, with a smooth surface. They consist of two cells and exhibit slight shrinkage at the diaphragm. The spore size was 20.9–27.3 (23.5) × 11.9–14.4 (12.8) μm (Figure 2, Table 2). The morphological characteristics of GUCC220045 resembled the description of *A. oligospora* by Ocampo-Gutierrez et al. [37].

*A. sinensis* (GUCC220046) soil samples were collected from Huang Naitang Town, Dafang County, Bijie City, Guizhou Province. The purification method was similar to that used for the previous strains. The mycelium spread all over the 9 cm Petri dish after about 7 days of incubation. The mycelium was sparse and had a cottony texture. The colonies were initially white, later producing a yellow pigment and turning light yellow. The backside of the colonies was yellow. The nutrient mycelium was septate with branching. The conidiophores were transparent, grew upright, and were rarely branched, with each conidiophore producing only one spore. The conidia were transparent, with 1–3 septa, and some conidia consisted of 4 cells. The middle cell was expanded, gradually narrowing towards the ends, with the head forming a papilla. The spore length and middle width were 20.5–33.1 (25.9) × 12.1–17.3 (14.4) μm (Figure 3, Table 2). The morphological characteristics of GUCC220046 were similar to the description of *A. sinensis* by Hastuti et al. [38].

The ITS sequence acquired in this study was aligned and submitted to GenBank for comparison with other sequences. The BLAST program revealed that the three isolates were part of the *Arthrobotrys* genus.

The phylogenetic tree inferred from the combined ITS and TEF dataset showed that the three isolated species clustered under the *Arthrobotrys* genus. Among these species, GUCC220044 and *A. megalosporus*, GUCC220045 and *A. oligospora*, and GUCC220046 and *A. sinensis* converged on a single branch (Figure 4). The supports for these classifications were 99%, 100%, and 79%, respectively, which indicates relative stability.

We inferred that these three strains belong to known species, and combined with the morphological results, these three strains were identified as *A. megalosporus*, *A. oligospora*, and *A. sinensis*.

### 3.2. Evaluation of Fungal Predatory Activity vs. Plant Parasitic Nematodes

In vitro screening of nematophagous fungi is a crucial method for assessing their potential for controlling phytonematodes before applying them for biological control. When examining the WA isolates under a microscope, no trap formation was observed. However, when the nematodes came into contact with the traps, they adhered to the trap bundles and could not move freely. The traps secreted chemicals that caused the nematodes to enter a comatose state and stop struggling, while enzymes decomposed the nematodes’ body walls. The mycelia invaded the nematodes and grew within them until they were entirely decomposed (Figure 5, Figure 6 and Figure 7). The number of traps and the efficiency of predatory nematodes increased over time (Figure 5, Figure 6 and Figure 7).

To understand whether there were differences in the predation effects of the tested strains on nematodes in different living environments, we evaluated their predatory effects on three different parasitic plant nematodes: *A. besseyi*, *B. xylophilus*, and *D. destructor*. The experimental results indicated that, based on the nematode species, there were differences in the production time and predation efficiency of the strains’ predatory organs.

At 48 h, the predation efficiencies of *A. megalosporus* were 99.7%, 83%, and 21.1%; for *A. oligospora,* they were 97.3%, 97.3%, and 54.6%; and for *A. sinensis*, they were 85.1%, 68.3%, and 19% for the three nematode species, respectively. The three strains produced the shortest time and the highest predation rates when they encountered rice acanthium, and produced the fewest predators and the lowest predation rates when they encountered the rotten stem nematode (Table 3).

## 4. Discussion

Based on the results of the phylogenetic tree, all three isolates belonged to known species. However, when referring to the corresponding strain size data listed in Table 3, it is evident that while the conidia morphology and size of *A. oligospora* and *A. sinensis* in this study were similar, there were some noteworthy differences. The strains studied here had slightly larger conidia than those previously reported [37,38]. For *A. megalosporus*, there are fewer relevant literature records of this nematode-eating fungus. According to the description of this fungus by Kano, the conidia are prismatic, oblong, or ovoid, and their size is 40–75 × 18–35 μm with 2–5 septa, and each conidiophore top produces one conidium. However, no images of the morphological characteristics of related strains are shown in the article. There were differences in the sizes of the conidia observed in the present study, as most of the conidia of *A. megalosporus* were smaller than those described in the article. It was also observed that one conidiophore could produce 1–3 conidia, which was not mentioned in the previous report. The reason for this phenomenon is that there are some morphological differences between strains of this species from different sources. This observation is supplemented by the morphological descriptions.

According to the results of the multi-gene phylogenetic tree constructed in this study, *A. megalosporus* and *A. reticulatus* are not grouped together, indicating a genetic divergence between the two species. Additionally, the existence of these two distinct species is substantiated by the relevant literature [34,39]. The GUCC220044 and *A. megalosporus* (TWF800) strains clustered into a branch with 99% similarity in the multi-gene phylogenetic tree; combined with its morphological characteristics, we identified it as *A. megalosporus*. The latest name for *A. oligosporus* (Syn. *A. oligospora*) in the Index Fungorum and Mycobank database is *Orbilia oligospora*. The names *O. oligospora* and *A. oligosporus* refer to the sexual and asexual stages of the same species. Although the name of the species has been updated in different databases, some scholars who specialize in nematode-trapping fungi still choose to use the name of *A. oligospora*, so we chose this name [39]. The morphological differences between *A. sinensis* and *A. thaumasius* are minimal, with only subtle variations observed in spore shape and size range [17,40]. The phylogenetic tree analysis in our study also demonstrated that these two nematode-trapping fungi belonged to distinct species, which aligned with the findings reported by Zhang et al. [39]. Based on morphological and phylogenetic tree analysis, the GUCC220046 strain was identified as *A. sinensis.*

In vitro experiments were performed to evaluate the nematicidal activity of local predatory fungal isolates. The combined cultivation of predatory fungi and parasitic nematodes on an agar medium allowed us to identify active strains of *Arthrobotrys* that formed traps of varying complexity, from simple rings to complex three-dimensional traps. As shown by the microscopic examination, all types of traps were capable of catching nematodes upon contact. The longer the contact time, the more the trap that formed. Interestingly, previous research has shown varying results with other plant or non-plant nematodes. For instance, *A. oligospora* showed a 45% predation rate for *B. xylophilus* at 25 °C for 48 h in lack corn-meat-agar (LCMA) media and up to a 74% predation rate for root-knot nematodes [41]. *A. megalosporum* has shown selectivity in its predation, preying only on pine wood nematodes [36]. Our experiment evaluated the predation abilities of the three isolates on three plant nematodes with different parasitic modes: *A. besseyi*, *B. xylophilus*, and *D. destructor*. These nematodes have varied hosts and effects, with *B. xylophilus* being a rapid spreader that threatens the pine forest ecosystem, *D. destructor* damaging underground parts of plants such as tubers and bulbs, and the rice dry-point nematode affecting rice yields by parasitizing leaves or grains.

The experimental results showed the potential of these three fungal strains in the biological control of plant nematode diseases. However, the predation efficiency varied across the nematodes tested. The predation rates were highest for *A. besseyi*, while the rates for *D. destructor* were not high, particularly for *A. megalosporus* and *A. sinensis*. This could be due to differences in the mutual attraction between different nematodes and the isolated strains. For instance, the strains might not produce enough sticky traps in response to the chemicals released by the nematodes, which might reduce the probability of nematodes being captured. From the perspective of nematode activity, the activity of *D. destructor* was lower than that of the other two nematodes, and their active direction was mostly not in the mycelium layer but down in the agar layer. Based on the above two phenomena, it is speculated that the contact frequency between *D. destructor* nematodes and hyphae was lower than that of the other two nematodes, which leads to a decrease in the probability or speed of hyphae specialization into predatory rings, so the predatory efficiency toward *D. destructor* is much lower than that of the other two nematodes at the same time point. There could also be variations in the body wall cuticle of the nematodes that affect the predation rates. In terms of frequency of isolation, *A. oligospora* was found to be ubiquitous in soil, while *A. megalosporus* and *A. sinensis* were less frequently isolated. These results have shown that an increase in fungal hyphal body size goes hand in hand with the fungal ability to trap and assimilate prey.

## Figures and Tables

**Figure 1 jof-09-01125-f001:**
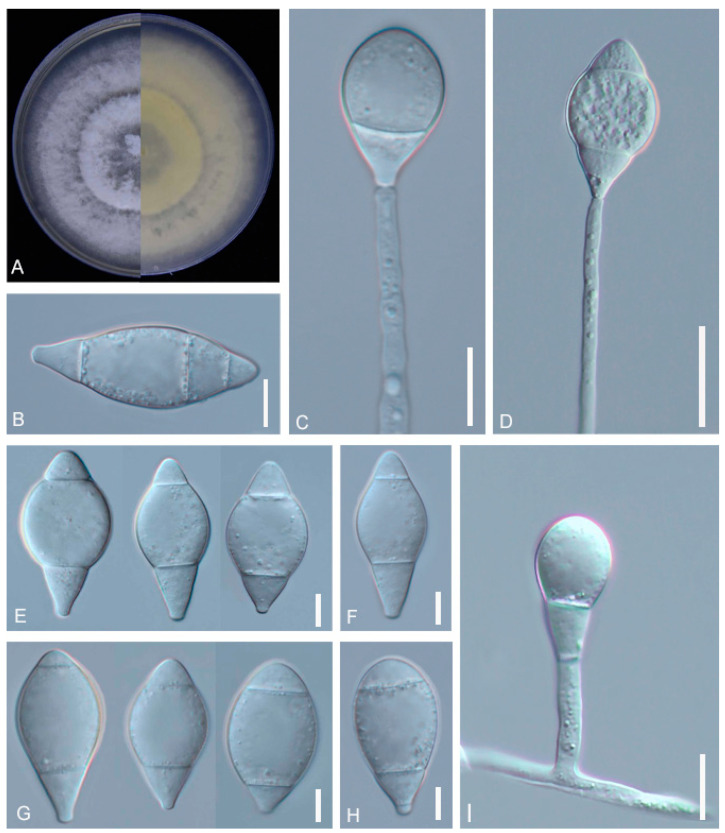
Colony and microscopic features of *A. megalosporus* (GUCC220044). (**A**) Colony. (**B**,**E**–**H**) Conidia. (**C**,**D**,**I**) Conidiophore. Scale bars: (**B**,**C**,**E**–**H**) = 10 μm; (**D**,**I**) = 20 μm.

**Figure 2 jof-09-01125-f002:**
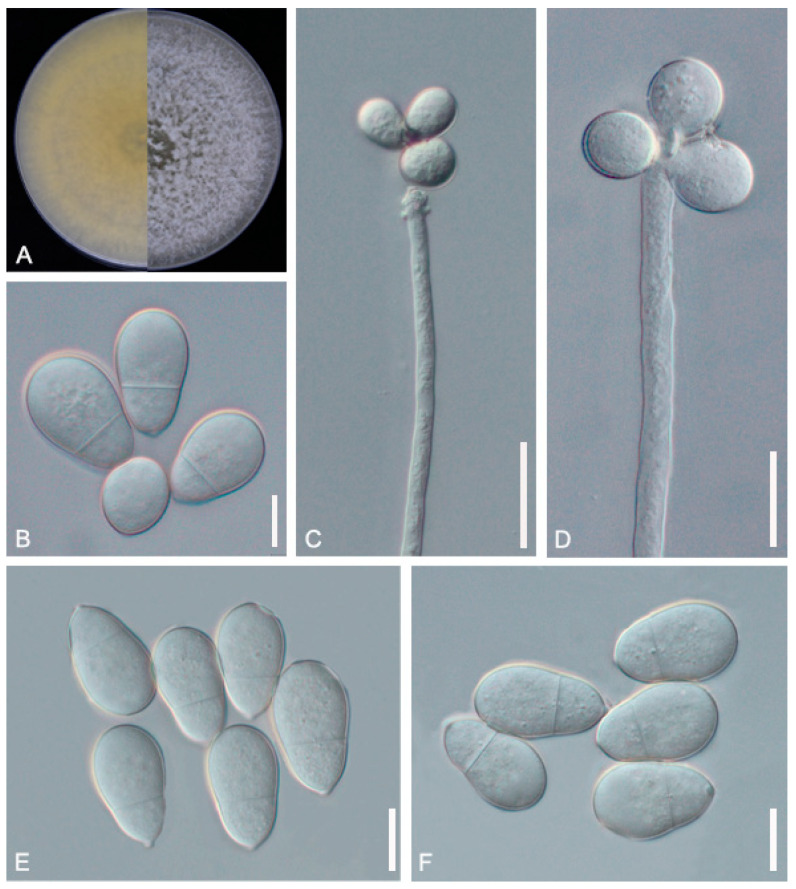
Colony and microscopic features of *A. oligospora* (GUCC220045). (**A**) Colony. (**B**,**E**,**F**) Conidia. (**C**,**D**) Conidiophore. Scale bars: (**B**,**D**–**F**) = 10 μm; (**C**) = 20 μm.

**Figure 3 jof-09-01125-f003:**
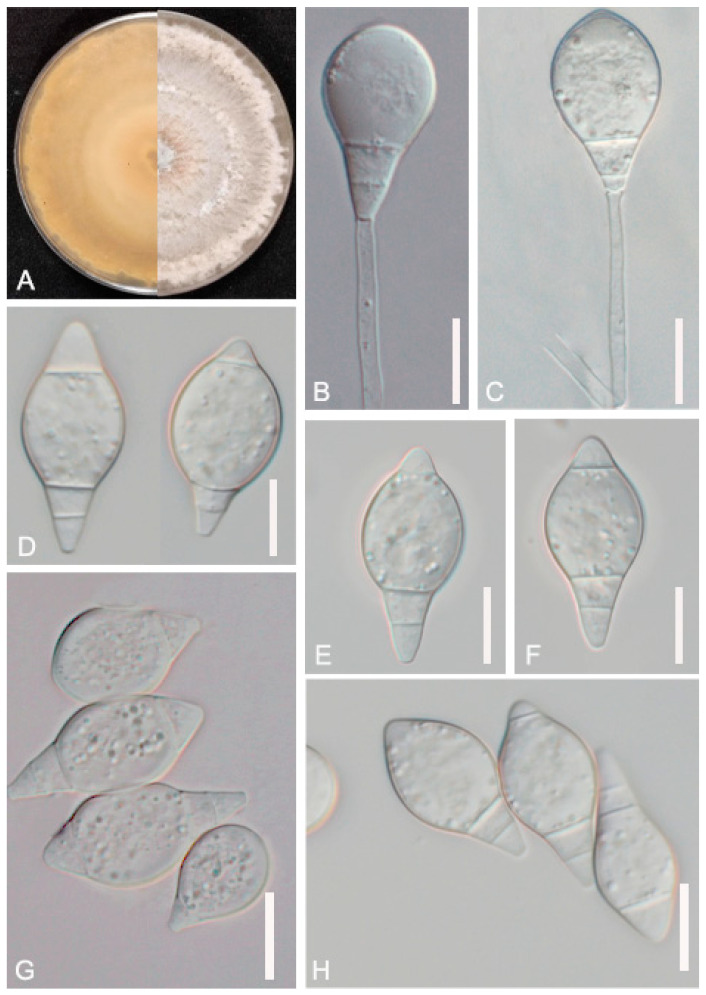
Colony and microscopic features of *A. sinensis* (GUCC220046). (**A**) Colony. (**D**,**E**–**H**) Conidia. (**B**,**C**) Conidiophore. Scale bars: (**B**–**H**) = 10 μm.

**Figure 4 jof-09-01125-f004:**
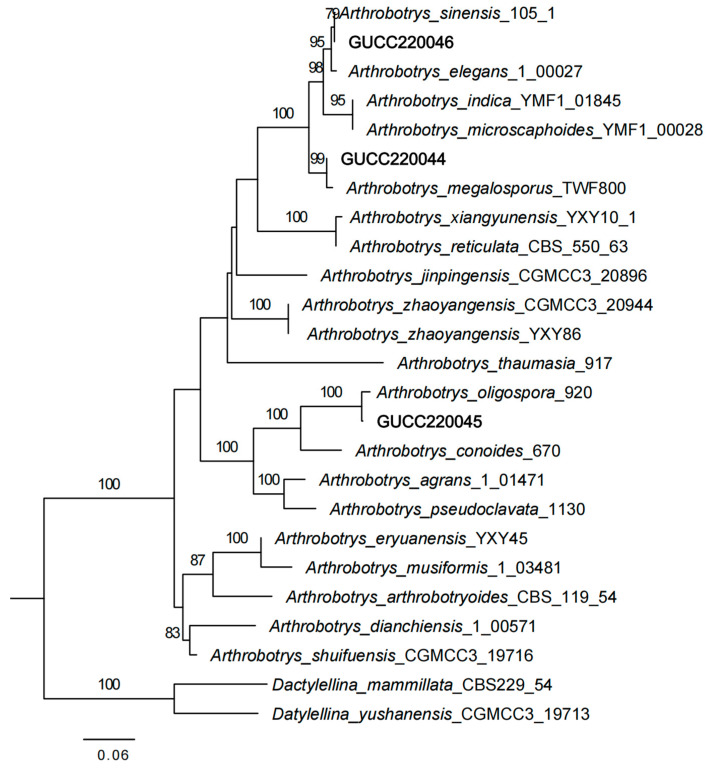
Maximum likelihood tree based on combined ITS and TEF sequences from 22 species of Orbiliaceae nematode-trapping fungi. The numbers on the tree branches indicate the bootstrap values from 1000 replications and an RAxML bootstrap support rate of ≥70%.

**Figure 5 jof-09-01125-f005:**
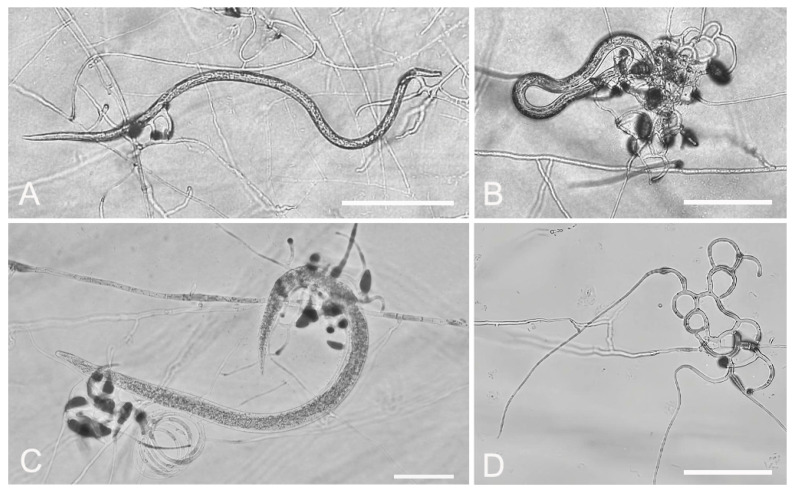
Predatory activity of *A. megalosporus* (GUCC220044) against three plant-parasitic nematodes on 2% WA. (**A**) Predatory activity against *A. besseyi*. (**B**) Predatory activity against *B. xylophilus*. (**C**) Predatory activity against *D. destructor*. (**D**) Trapping device. Scale bars: (**A**,**B**,**D**) = 200 μm; (**C**) = 50 μm.

**Figure 6 jof-09-01125-f006:**
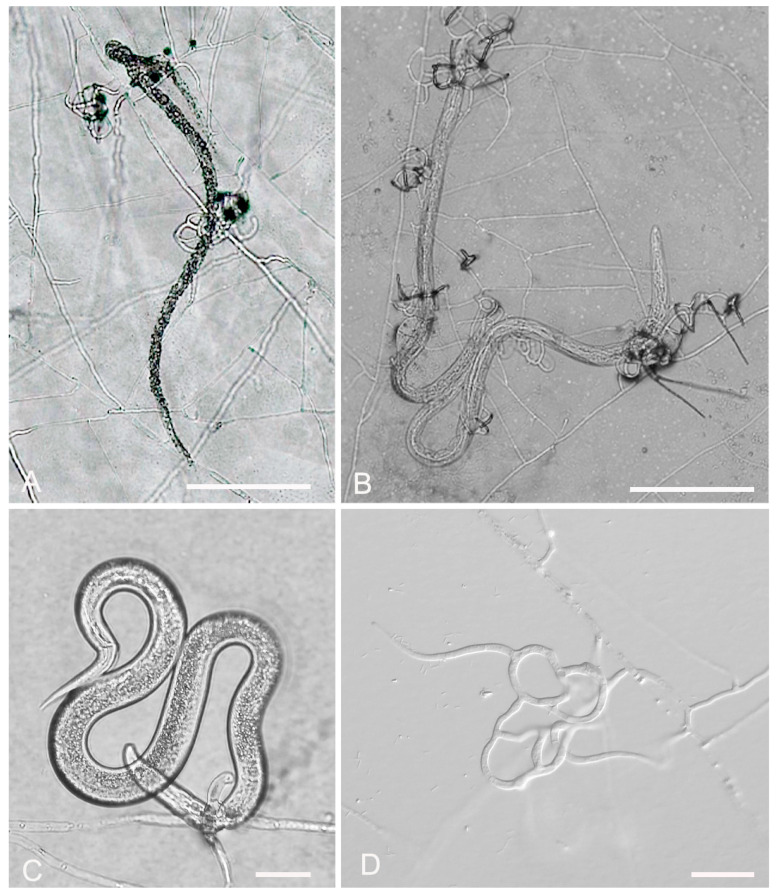
Predatory activity of *A. oligospora* (GUCC220045) against three plant-parasitic nematodes on 2% WA. (**A**) Predatory activity against *A. besseyi*. (**B**) Predatory activity against *B. xylophilus*. (**C**) Predatory activity against *D. destructor*. (**D**) Trapping device. Scale bars: (**A**,**B**) = 200 μm; (**C**) = 50 μm; (**D**) = 20 μm.

**Figure 7 jof-09-01125-f007:**
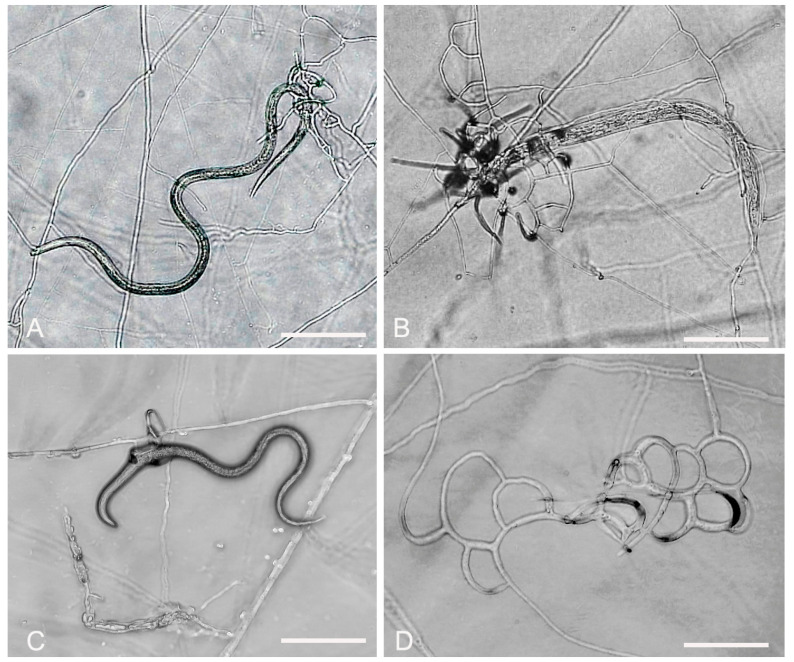
Predatory activity of *A. sinensis* (GUCC220046) against three plant-parasitic nematodes on 2% WA. (**A**) Predatory activity against *A. besseyi*. (**B**) Predatory activity against *B. xylophilus*. (**C**) Predatory activity against *D. destructor*. (**D**) Trapping device. Scale bars: (**A**,**B**) = 200 μm; (**C**) = 100 μm; (**D**) = 50 μm.

**Table 1 jof-09-01125-t001:** The GenBank accession numbers of the isolates included in this study.

Taxon	Strain Number	GenBank Accession Number		Reference
		ITS	TEF	
*A. arthrobotryoides*	CBS 119.54	MH857262	_	[33]
*A. conoides*	670	AY773455	AY773397	[32]
*A. dianchiensis*	1.00571	MH179720	_	[34]
*A. elegans*	1.00027	MH179688	_	Unpublished
*A. eryuanensis*	YXY45	ON808616	ON809547	[34]
*A. flagrans*	1.01471	MH179741	MH179583	[34]
*A. indica*	YMF1.01845	KT932086	_	[35]
*A. jinpingensis*	CGMCC3.20896	OM855569	OM850311	[34]
*A. megalosporus*	TWF800	MN013995	_	[34]
*A. microscaphoides*	YMF1.00028	MF948395	MF948552	Unpublished
*A. musiformis*	1.03481	MH179783	MH179607	[34]
*A. oligospora*	920	AY773462	AY773404	[32]
*A. pseudoclavata*	1130	AY773446	AY773388	[32]
*A. shuifuensis*	CGMCC3.19716	MT612334	OM850306	[34]
*A. sinensis*	105-1	AY773445	AY773387	[32]
*A. xiangyunensis*	YXY10-1	MK537299	_	[34]
*A. zhaoyangensis*	CGMCC3.20944	OM855568	OM850310	[34]
*A. zhaoyangensis*	YXY86	ON808620	ON809551	[34]
*A. thaumasia*	917	AY773461	AY773403	[32]
*A. reticulata*	CBS 550.63	MH858355	_	[32]
*Dactylellina mammillata*	CBS229.54	AY902794	DQ999843	[34]
*D. yushanensis*	CGMCC3.19713	MK372061	MN915113	Unpublished

**Table 2 jof-09-01125-t002:** Conidia dimensions and other taxonomic characteristics of *Arthrobotrys*.

Genus/Species	Conidia Size		Reference
	Mean (μm)	Range (μm)	
*A. megalosporus*	Length: —	40–75	[36]
	Width: —	18–35	
*A. megalosporus*	Length: 29.4	24.32–41.1	GUCC220044
	Width: 15	11.69–17.5	
*A. oligospora*	Length: 21.4	17.5–25.6	[37]
	Width: 11.8	11.4–12.5	
*A. oligospora*	Length: 23.5	20.92–27.3	GUCC220045
	Width: 12.8	11.88–14.4	
*A. sinense*	Length: 25.5	20.4–30	[38]
	Width: 20	18–22	
*A. sinense*	Length: 25.9	20.50–33.1	GUCC220046
	Width: 14.4	12.14–17.3	

**Table 3 jof-09-01125-t003:** The predation rates of three strains against three species of plant-pathogenic nematodes on water agar.

Nematode	Treatment	24 h Mortality (%)	48 h Mortality (%)
*A. besseyi*	*A. megalosporus*	83.3 ± 6.7 a	99.7 ± 0.7 a
	*A* *. oligospora*	84.5 ± 6.5 a	97.3 ± 0.9 a
	*A. sinensis*	69 ± 6.7 b	85.1 ± 5.5 b
	control	0	0
*B. xylophilus*	*A. megalosporus*	24 ± 3.2 b	83 ± 5.3 b
	*A* *. oligospora*	51.1 ± 2.7 a	97.3 ± 1.26 a
	*A. sinensis*	52.3 ± 1.3 a	68.3 ± 6.7 c
	control	0	0
*D. destructor*	*A. megalosporus*	3.4 ± 0.7 c	21.1 ± 4.3 b
	*A* *. oligospora*	23.8 ± 5.5 a	54.6 ± 5.0 a
	*A. sinensis*	10.5 ± 2.8 b	19.0 ± 0.7 ab
	control	0	0

Note: Data represent the mean ± SD (n = 4). Different lowercase letters indicate a significant difference (*p* < 0.05) between the control and the three strains within a species and within a given time point.

## Data Availability

Data are contained within the article.

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
