# Peer review of "Isolation, Identification, and Evaluation of the Predatory Activity of Chinese Arthrobotrys Species towards Economically Important Plant-Parasitic Nematodes"

_jof, 2023, doi:10.3390/jof9121125_

Round 1

Reviewer 1 Report

Comments and Suggestions for Authors

I reviewed the manuscript "Isolation, identification, and evaluation of predatory activity of 2 nematophagous fungal species of Arthrobotrys against three types of plant-parasitic nematodes" sent to the Journal of Fungi. This work is interesting and well-written. I detected some problems that I describe as follows:

According to Index Fungorum, the correct name is Arthrobotrys megalosporus (Drechsler) M. Scholler, Hagedorn & A. Rubner, but according to Mycobank, the correct name is Arthrobotrys reticulatus (Peach) M. Scholler, Hagedorn & A. Rubner, Sydowia 51 (1): 104 (1999) [MB#627016].

Similarly, according to Index Fungorum, the correct name for Arthrobotrys oligospora is Arthrobotris oligosporus, and according to Mycobank, it is Arthrobotrys aggregatus Mekht., Khishchnye nematofagovye Griby-Gifomitsety [Predacious nematophagous hyphomycetes]: 69 (1979) [MB#626946], but the synonymy is not clear.

Finally, according to Mycobank the current name of Arthrobotrys sinensis is Arthrobotrys thaumasius (Drechsler) S. Schenck, W.B. Kendr. & Pramer, Canadian Journal of Botany 55 (8): 984 (1977) [MB#627000]. The authors should correct and/or report the synonymy and analyze if the work with these strains belonging to this species is indeed a novelty.

Some Latin words are italicized, and others are not, for example, lines 14, 81, and 83.

The methodology explained between lines 87 and 95 includes the fungus Botrytis cinerea without proper citation; it is necessary to explain this and include a citation.

Lines between 165 and 170 in the Results section must be included in Materials and Methods.

In the descriptions, two decimals are used; it is not necessary to include more than one decimal.

In line 181, Arthrobotrys is not italicized; please review the entire text.

Author Response

Point 1: According to Index Fungorum, the correct name is Arthrobotrys megalosporus (Drechsler) M. Scholler, Hagedorn & A. Rubner, but according to Mycobank, the correct name is Arthrobotrys reticulatus (Peach) M. Scholler, Hagedorn & A. Rubner, Sydowia 51 (1): 104 (1999) [MB#627016].

Similarly, according to Index Fungorum, the correct name for Arthrobotrys oligospora is Arthrobotris oligosporus, and according to Mycobank, it is Arthrobotrys aggregatus Mekht., Khishchnye nematofagovye Griby-Gifomitsety [Predacious nematophagous hyphomycetes]: 69 (1979) [MB#626946], but the synonymy is not clear.

Finally, according to Mycobank the current name of Arthrobotrys sinensis is Arthrobotrys thaumasius (Drechsler) S. Schenck, W.B. Kendr. & Pramer, Canadian Journal of Botany 55 (8): 984 (1977) [MB#627000]. The authors should correct and/or report the synonymy and analyze if the work with these strains belonging to this species is indeed a novelty.

Response 1: Dear Reviewer, Thank you very much for your constructive feedback on our manuscript. We have revised the article according to your suggestion and made the following reply

We carefully compared the name differences of Arthrobotrys megalosporus in Index Fungorum, Mycobank. According to the results of the multi-gene phylogenetic tree constructed in the article, Arthrobotrys megalosporus and Arthrobotrys reticulatus are not clustered together, indicating that there is a genetic distance between the two. And in other literatures, these two species are also used as separate species to establish phylogenetic trees. So we use the name in Index Fungorum.

  1. Zhang, F.; Boonmee, S.; Bhat, J.D.; Xiao, W.; Yang, X.Y. New ArthrobotrysNematode-Trapping Species (Orbiliaceae) from Terrestrial Soils and Freshwater Sediments in China. J. Fungi 2022, 8, 671. 
  2. Zhang, F.; Boonmee, S.; Yang, Y.-Q.; Zhou, F.-P.; Xiao, W.; Yang, X.-Y. Arthrobotrys Blastospora Nov. (Orbiliomycetes): A Living Fossil Displaying Morphological Traits of Mesozoic Carnivorous Fungi. Journal of Fungi 2023, 9, 451

Arthrobotrys oligosporus is currently the latest name in the Index Fungorum is Orbilia oligospora, Orbilia oligospora and Arthrobotrys oligosporus are the same species of sexual and asexual type, “in Baral & E. Weber, in Baral, Weber & Marson. In Monogr. Orbiliomycetes : 1539 ( 2020 ),” However, in the latest taxonomic literature, the author still uses the name of the asexual species, so we chose the name Arthrobotrys oligospora, which we all use.

Zhang, F.; Boonmee, S.; Yang, Y.-Q.; Zhou, F.-P.; Xiao, W.; Yang, X.-Y. Arthrobotrys Blastospora Sp. Nov. (Orbiliomycetes): A Living Fossil Displaying Morphological Traits of Mesozoic Carnivorous Fungi. Journal of Fungi 2023, 9, 451

Arthrobotrys sinensis and Arthrobotrys thaumasius are two species in Index Fungorum. According to the literature description, the difference in shape between the two species is small, and the spore size is different. However, in the phylogenetic tree, these two species are not clustered together, indicating that there is a genetic distance between the two. In the literature in recent years, they also do two species to compare. For example

  1. Hastuti, L.; Berliani, K.; Budi Mulya, M.; Hartanto, A.; Pahlevi, S.Arthrobotrys Sinensis (Orbiliaceae Orbiliales), a New Record of Nematode-Trapping Fungal Species for Sumatra, Indonesia (2022).
  2. Doolotkeldieva, T.; Bobushova, S.; Muratbekova, A.; Schuster, C. Leclerque A. Isolation, Identification, and Characterization of the Nematophagous Fungus Arthrobotrys oligosporafrom Kyrgyzstan. Acta Parasitol 2021

Point 2: Some Latin words are italicized, and others are not, for example, lines 14, 81, and 83. In line 181, Arthrobotrys is not italicized; please review the entire text.

Response 2: We sincerely apologize for the issues related to grammar and typing errors that you encountered in our initial submission, we have modified the italic error in the text. We hope that this comprehensive editing has significantly improved the quality of our manuscript, and we appreciate your patience and understanding. We look forward to your further comments and suggestions.

Point 3: The methodology explained between lines 87 and 95 includes the fungus Botrytis cinerea without proper citation; it is necessary to explain this and include a citation.

Response 3: We accept your comments. We added the corresponding literature after Botrytis cinerea.

Point4: Lines between 165 and 170 in the Results section must be included in Materials and Methods.

Respons 4: Thank you for your valuable comments, we have added the content between 165-170 to the method and material according to yours. We have added the modified content to lines 115-122.

Point5: In the descriptions, two decimals are used; it is not necessary to include more than one decimal.

Respons 5: Thank you for your comments. According to your suggestion, we will change the description data in the text to keep only one decimal processing.

Point6: In line 181, Arthrobotrys is not italicized; please review the entire text.

Respons 6: Thank you very much!We We have modified.

Reviewer 2 Report

Comments and Suggestions for Authors

Major revision.

The research is actual and useful as the contribution to biological control technology against nematode pests.

The sample size (n=3) for the predation rate count of three fungal strains is too low, I would also recommend to use SD for statistics and not SE.

Other notes:

L28

 and fruits such as rice, corn, tomatoes, beets, and 28 kiwifruit

= rice, corn and beets are not fruits!!

L 178

morphological 178 characteristics were consistent with the description of A. megalospora by Kano et al., but 179 there are some differences in spore size.

= include the year of Kano et al.

L205

Petri dish or petri dish? = please use the name uniformly in the text

L214

 Please use A. sinensis in italics font

L235 In vitro predatory activity of identification fungal against nematodes

= Please re-phrase to make the sentence understandable

L274-277 (Table 3) Note: Data represent the mean±SE (n=3). Different lowercase letters indicate a signifcant difference 276 (P<0.05) between the control and three strains within a species and within a given time point

Why you use only n=3 for each measuring value? The sample (n=3) is low. And why you use SE and not the SD (standard deviation), SD is most adequate for the the same type of the statistical distribution using the same isolate of the nematode species.

L 293-299

= this section is a repetition of the information which already done in the Introduction

L321

For instance, the strains might not produce enough sticky bacterial rings

Bacterial rings in mycelium? Please explain

Author Response

Point 1: The sample size (n=3) for the predation rate count of three fungal strains is too low, I would also recommend to use SD for statistics and not SE.

Response 1: Dear Reviewer, thank you very much for your constructive feedback on our manuscript. We agree with you that we supplemented the predation experiment and added repeated processing, and used SD data to reanalyze to ensure the reliability and authenticity of the experiment. The reprocessed data has been placed in Table 3 of this article.

Point 2: L28 and fruits such as rice, corn, tomatoes, beets, and 28 kiwifruit= rice, corn and beets are not fruits!!

Response 2 :Thank you for your comments, we have removed redundant duplicate content.

Point 3: L 178 morphological 178 characteristics were consistent with the description of A. megalospora by Kano et al., but 179 there are some differences in spore size.

= include the year of Kano et al.

Response 3: Thank you for your comments, we measured the spore size again based on your comments, and the measurement range of spore size was unchanged. The reason for this phenomenon may be that the source of the strain is different, so it may be different from the size of the strain described in this paper. We also explained it in the discussion.. Line292-293

Point 4: L205 Petri dish or petri dish? = please use the name uniformly in the text

Response 4: We agree with you that the petri dish format has been unified.

Point 5: L214  Please use A. sinensis in italics font

Response 5: I 'm sorry for the format error in this article, we have modified the relevant content.

 Point 6: L235 In vitro predatory activity of identification fungal against nematodes

= Please re-phrase to make the sentence understandable

Response 6: We agree with your opinion that we have re-made sentences. Line 235

 Point 7: L274-277 (Table 3) Note: Data represent the mean±SE (n=3). Different lowercase letters indicate a signifcant difference 276 (P<0.05) between the control and three strains within a species and within a given time point

Why you use only n=3 for each measuring value? The sample (n=3) is low. And why you use SE and not the SD (standard deviation), SD is most adequate for the the same type of the statistical distribution using the same isolate of the nematode species.

Response 7: Thank you for your valuable comments, we have increased the experiment repeated processing, and use SD for data analysis, to ensure the reliability and authenticity of the experiment. The reprocessed data has been placed in Table 3 of this article.

Point 8: L 293-299 = this section is a repetition of the information which already done in the Introduction

Response 8: I agree with you that we have removed the content in L 293-299 and re-edited it for contextual cohesion, which should allow readers to more effectively grasp the purpose of our research without getting lost in repetitive information. Line 295-301

 Point 9: L321 For instance, the strains might not produce enough sticky bacterial rings

Bacterial rings in mycelium? Please explain

Response 9: I 'm sorry for this low-level error in the article, the correct term should be “strap” or :”ring”. We correct it in the text and explain this phenomenon. For example, the activity frequency of Ditylenchus destructor is low, or the activity inside the medium. LIine 318-325

Round 2

Reviewer 1 Report

Comments and Suggestions for Authors

The manuscript has been significantly improved

I understand that the authors follow the names according to the results of the phylogenies, but they should discuss why they choose the name they use in the text (e.g. Arthrobotrys megalosporus) as well, beyond answering the reviewer. They should reflect the reasoning in the manuscript to enrich it.

Author Response

Thank you very much for your constructive suggestions. We have revised the article according to your suggestion. The relevant content has been incorporated into the discussion to provide explanation.